# Ovarian Cancer in the Era of Immune Checkpoint Inhibitors: State of the Art and Future Perspectives

**DOI:** 10.3390/cancers13174438

**Published:** 2021-09-03

**Authors:** Brigida Anna Maiorano, Mauro Francesco Pio Maiorano, Domenica Lorusso, Evaristo Maiello

**Affiliations:** 1Oncology Unit, Foundation Casa Sollievo della Sofferenza IRCCS, 71013 San Giovanni Rotondo, Italy; e.maiello@operapadrepio.it; 2Department of Translational Medicine and Surgery, Catholic University of the Sacred Heart, 00168 Rome, Italy; 3Division of Obstetrics and Gynecology, Biomedical and Human Oncological Science, University of Bari “Aldo Moro”, 70121 Bari, Italy; m.maiorano23@studenti.uniba.it; 4Gynecologic Oncology Unit, Catholic University of the Sacred Heart, 00168 Rome, Italy; domenica.lorusso@policlinicogemelli.it; 5Scientific Directorate, Fondazione Policlinico “A.Gemelli” IRCCS, 00168 Rome, Italy

**Keywords:** ovarian cancer, checkpoint inhibitors, ICIs, immunotherapy, PARP, avelumab, pembrolizumab, nivolumab, bevacizumab, platinum

## Abstract

**Simple Summary:**

Ovarian cancer (OC) represents the fifth leading cause of cancer-related deaths among women. In the advanced disease setting, OC recurrence after chemotherapy is over 70% in the first 2 years, with few therapeutic options. Immunotherapy with the immune checkpoint inhibitors (ICIs) showed high efficacy and changed the therapeutic scenario of many tumors in the last 10 years. With our review, we aimed to summarize the clinical trials of ICIs in OC. In OC, ICIs clinical trials have reported poor outcomes in terms of patient response and survival, with some studies failing to reach their objectives. Combining immunotherapy with drugs targeting different pathways might enhance efficacy and overcome cancer resistance. The search for biomarkers predicting ICIs response is essential for the identification of patients most likely to benefit from ICI therapy.

**Abstract:**

Background: Ovarian cancer (OC) represents the eighth most common cancer and the fifth leading cause of cancer-related deaths among the female population. In an advanced setting, chemotherapy represents the first-choice treatment, despite a high recurrence rate. In the last ten years, immunotherapy based on immune checkpoint inhibitors (ICIs) has profoundly modified the therapeutic scenario of many solid tumors. We sought to summarize the main findings regarding the clinical use of ICIs in OC. Methods: We searched PubMed, Embase, and Cochrane Databases, and conference abstracts from international congresses (such as ASCO, ESMO, SGO) for clinical trials, focusing on ICIs both as monotherapy and as combinations in the advanced OC. Results: 20 studies were identified, of which 16 were phase I or II and 4 phase III trials. These trials used ICIs targeting PD1 (nivolumab, pembrolizumab), PD-L1 (avelumab, aterolizumab, durvalumab), and CTLA4 (ipilimumab, tremelimumab). There was no reported improvement in survival, and some trials were terminated early due to toxicity or lack of response. Combining ICIs with chemotherapy, anti-VEGF therapy, or PARP inhibitors improved response rates and survival in spite of a worse safety profile. Conclusions: The identification of biomarkers with a predictive role for ICIs’ efficacy is mandatory. Moreover, genomic and immune profiling of OC might lead to better treatment options and facilitate the design of tailored trials.

## 1. Introduction

Ovarian cancer (OC) accounts for about 2% of tumors, representing the eighth most common cancer among the female population. The incidence is around 11 cases/100,000 inhabitants/year, and it is higher among white women [1,2]. The frequency of OC rises with age, being uncommon before 30, and more frequently presenting at 50–70. Globally, ovarian cancer represents the fifth leading cause of female cancer-related deaths, with a 5 y survival rate falling from 90% at stage I to 25% at stage IV [2]. The majority of OCs have an epithelial origin, among whom serous carcinoma has the most aggressive features and is usually diagnosed at advanced stages [3]. Platinum-based chemotherapy regimens represent the mainstay of treatment [4,5,6,7]. The response to these agents and the treatment-free interval (TFI) after platinum define the subsequent treatment, moving from the platinum-refractory (PR) (relapse < 6 months from the platinum end) to the platinum-sensitive (PS) patients (TFI > 12 mos). Despite initial benefits, disease recurrence occurs in over 2/3 of patients within the first two years. Therefore, new drugs were explored, and other agents such as the PARP-inhibitor (PARPi) agents and the anti-vascular endothelial growth factor (VEGF) bevacizumab were approved in the advanced setting [8,9].

Immunotherapy has represented a breakthrough therapy for many solid tumors [10]. Thus far, the best-studied mechanisms for inducing an immune response against tumors rely on inhibiting the immune checkpoint. The immune checkpoint inhibitors (ICIs) consist of monoclonal antibodies targeting Programmed Cell Death Protein 1 (PD-1)/Programmed Death-Ligand 1 (PD-L1) or Cytotoxic T-Lymphocyte Antigen 4 (CTLA-4), expressed by tumor or immune cells. After binding with these ligands, ICIs remove the inhibition signals for the immune system, unlocking the anti-tumor response [11]. However, in OC, ICIs reported modest results, and some phase III trials were prematurely terminated for futility. Combinations with other compounds, such as PARPis or anti-angiogenic drugs, represent promising opportunities to enhance the clinical effectiveness of immunotherapy [12,13,14,15,16,17,18,19,20,21,22,23,24,25,26,27,28,29,30,31,32,33,34,35,36].

We hereinafter ought to synthesize the clinical trials involving ICIs that were conducted in advanced OC to discuss the pros and cons and explore future perspectives to maximize the efficacy of immunotherapy for most women with advanced disease.

## 2. Materials and Methods

We searched the PubMed, EMBASE, and Cochrane databases and abstracts from international conferences (e.g., ASCO, ESMO, SGO). The terms (‘ovarian cancer’ OR ‘ovarian carcinoma’) AND (‘immune checkpoint inhibitor’ OR ICI OR avelumab OR nivolumab OR atezolizumab OR pembrolizumab OR durvalumab OR tremelimumab OR ipilimumab OR ‘anti PD1’ OR ‘anti PD-L1’ OR ‘anti CTLA4’) were used. Papers published in peer-reviewed journals and conference abstracts in the English language up to June 2021 were selected. We included clinical trials, whereas reviews, letters, and personal opinions were excluded. A total of 20 studies were included in our review.

## 3. Results

Since the FDA approved ipilimumab for advanced melanoma in 2011, the last ten years were characterized by the widespread use of ICIs, revolutionizing the therapeutic algorithm for many solid tumors [10,37]. ICIs are monoclonal antibodies that promote the anti-tumoral response of the host immune system through the inhibition of negative signals for effector T-cells. Among the ICIs tested in clinical trials in OC, pembrolizumab and nivolumab target PD-1, atezolizumab, avelumab, and durvalumab bound PD-L1, ipilimumab and tremelimumab are directed against CTLA-4. Sixteen phase I or II and 4 phase III trials were published (Table 1).

### 3.1. Anti PD1 Agents

Nivolumab and Pembrolizumab were tested as single agents and combined with other ICIs, chemotherapy, anti-angiogenic agents, and PARPis.

#### 3.1.1. Pembrolizumab

In the KEYNOTE-100 (NCT02674061) phase II study, pembrolizumab 200 mg q3w was administered to two cohorts of patients with recurrent ovarian cancer (ROC): cohort A enrolled 285 patients after one to three prior therapies with a treatment-free interval (TFI) of 3–12 months; cohort B included 91 progressive patients with up to six previous lines of therapy with a TFI of at least 3 mos. The primary endpoint was overall response rate (ORR) by Response Evaluation Criteria in Solid Tumors (RECIST) criteria and according to PD-L1 expression. Secondary endpoints included: duration of response (DoR), disease-control rate (DCR), progression-free survival (PFS), overall survival (OS), and safety. The combined ORR of the two cohorts was 8.0%, the overall DCR 37%, and around 1/3 of responses lasted more than 6 months. The mDoR was 8.2 mos in cohort A and not reached in cohort B. The mPFS was 2.1 mos [12]. PD-L1 positive patients (defined as a combined positive score-[CPS] ≥ 10) reached better results than PD-L1 negative, in terms of both ORR (17.1%) and mOS (21.9 mos-cohort A, and 24.0 mos-cohort B) [12,13]. The most common adverse events (AEs) were fatigue (33.8%), nausea (15.4%), and decreased appetite (10.6%), with 19.7% of women experiencing >G3 AEs. The most common immune-related AEs (irAEs) were thyroid disorders (17.5%). Two treatment-related deaths were recorded, and 5.1% of patients discontinued the treatment due to toxicity [12].

In the Keynote-028 (NCT02054806) multi-cohort phase Ib trial, only PD-L1 positive patients were included. Twenty-six women were treated with pembrolizumab in the OC cohort. The ORR represented the primary endpoint. After a median follow-up of 15.4 mos, ORR was 11.5%, mPFS 1.9 mos, and mOS was 13.8 mos. A total of 73.1% of patients experienced at least one treatment-related adverse event (TRAE): arthralgia (19.2%), nausea (15.4%), and pruritus (15.4%) were the most common. One G3 hypertransaminasemia was recorded, while no deaths or treatment discontinuation for toxicity occurred [14].

Attempts to combine pembrolizumab were made with chemotherapy, bevacizumab, or PARPis. Pembrolizumab plus liposomal doxorubicin (PLD) resulted in an ORR of 19%, 3 PR, and 1 SD >24 weeks, among 26 platinum-resistant patients, with no G4 or G5 toxicities. G2 pneumonitis occurred in 8% of patients. G3 AEs included rash (19%) and ALT increase (8%) [15]. In the NCT02853318 phase II trial, 40 platinum-progressive OC women were treated with pembrolizumab plus bevacizumab plus oral cyclophosphamide. The study met its primary endpoints, reaching an ORR of 47.5% and an mPFS of 10 mos. The 6-month PFS rate was 100% for the PS-ROC and 59% for the PR-ROC patients (*p* = 0.024). The most frequent AEs were fatigue (45.0%), diarrhea (32.5%), and hypertension (27.5%), while the most common ≥G3 AEs were hypertension (15.0%) and lymphopenia (7.5%) [16]. The TOPACIO/Keynote-162 (NCT02657889) was a phase I/II study evaluating the combination of pembrolizumab plus the PARPi niraparib, conducted among triple-negative breast cancer and ROC patients. In the PR-ROC cohort, 62 patients were treated. The ORR (primary endpoint of the study) was 25%, the DCR was 68%. In the breast cancer gene (BRCA)-mutant population, ORR and DCR were 45% and 73%, respectively. The most frequent ≥G3 TRAEs were anemia (21%) and thrombocytopenia (9%). No treatment-related deaths were recorded [17].

#### 3.1.2. Nivolumab

As a single agent, nivolumab was administered to 20 patients with PR-ROC in the UMIN000005714 phase II trial, evaluating the best overall response (BOR) as a primary endpoint: two complete responses (CR) were recorded, the DCR was 45%, the mPFS 3.5 mos, and the mOS 20.0 mos. Of note, ≥G3 TRAEs occurred in 40% of patients. Two patients (10%) experienced serious TRAEs, and 11% of patients discontinued Nivolumab treatment mainly due to treatment-related thyroid disorders [18].

In the NRG-GY003 phase II study (NCT02498600), nivolumab alone or plus ipilimumab was administered to 100 platinum-progressing ROC patients. The primary endpoint was ORR; secondary endpoints included PFS and OS, stratified by the platinum-free interval (PFI). The combination of nivolumab and ipilimumab vs. nivolumab alone resulted in increased ORR (31.4% vs. 12.2%; (*p* = 0.034), longer PFS (3.9 vs. 2 mos; HR = 0.53, 95% CI 0.34–0.82) and longer OS (28.1 vs. 21.8 mos; HR = 0.79, 95% CI 0.44–1.42). However, the combination treatment was less tolerated (>G3 AEs 49% vs. 33%). The response was not associated with PD-L1 status [19].

The combination of nivolumab plus bevacizumab was tested in 38 platinum-progressing epithelial ovarian cancer (EOC) patients in the NCT02873962 phase II study. ORR was the primary endpoint, while secondary endpoints were ORR according to platinum sensitivity and PD-L1 expression, PFS, and safety. The combination of nivolumab plus bevacizumab resulted in an ORR of 28.9%, ranging from 16.7% in the platinum-resistant (*n* = 18) to 40.0% in the platinum-sensitive patients (*n* = 20). Median PFS was 9.4 mos and 12.1 mos in the overall and platinum-sensitive population, respectively. Of note, better response rates were observed in patients with PD-L1 negative than PD-L1 positive disease. A total of 89.5% of patients developed AEs, of whom the most common were fatigue (47.4%), headache (28.9%), myalgia (28.9%), serum amylases increase (28.9%), aspartate aminotransferase level increase (26.3%), hypertension (26.3%). Four pneumonitis (10.5%) and two colitis (5.3%) cases were reported [20].

A unique phase III trial (NINJA) was conducted in the Japanese population, randomizing patients with PR-ROC to nivolumab (*n* = 157) versus gemcitabine or PLD (*n* = 159) at the physician’s choice. However, the trial failed its primary endpoint, as there was no difference between the two groups for OS (HR = 1.03, 95% CI, 0.8–1.32; *p* = 0.8). Moreover, mPFS was shorter in the nivolumab group (2.04 mos) than in the gem/PLD group (3.84 mos; HR = 1.46, 95% CI, 1.15–1.85; *p* = 0.002) The incidence of ≥G3 AEs was 22.4% with nivolumab and 68.4% with gem/PLD [21].

### 3.2. Anti-PD-L1 Agents

#### 3.2.1. Avelumab

In the OC cohort of the JAVELIN (NCT01772004) phase Ib study, avelumab 10 mg/kg q2w determined an objective response in 12/125 patients, including 1 CR and 11 partial responses (PR). The 1 y PFS rate was 10.2% (95% CI, 5.4–16.7%), the mPFS was 2.6 months (95% CI, 1.4–2.8 mos), and the mOS was 11.2 months (95% CI, 8.7–15.4 mos). The responses were recorded independently from PD-L1 expression. The most frequent AEs were fatigue (13.6%), diarrhea (12.0%), and nausea (11.2%). ≥G3 AEs occurred in 7.2% of patients, among which the most frequent was the increase in lipase level (2.4%) [22].

The combination of avelumab plus chemotherapy was tested in two randomized phase III trials. The JAVELIN Ovarian 100 (NCT02718417) evaluated carboplatin-paclitaxel chemotherapy alone versus chemotherapy plus avelumab followed by avelumab maintenance versus chemotherapy plus avelumab in the front line OC setting. Nine hundred and ninety-eight stage III-IV patients were enrolled. The primary endpoint was PFS. However, after a median follow-up of 11 mos, PFS was not improved in both avelumab arms, and the trial was stopped after meeting futility criteria [23]. Based on the absence of benefit from Avelumab in unselected patients, the JAVELIN Ovarian PARP 100 (NCT03642132) phase III study, with three arms consisting of chemotherapy plus avelumab followed by maintenance with avelumab plus talazoparib, chemotherapy followed by talazoparib maintenance, and chemotherapy plus bevacizumab followed by bevacizumab, was also terminated [24].

In the JAVELIN 200 (NCT02580058) phase III study, 566 patients with PR-ROC were randomized 1:1:1 to avelumab alone or avelumab plus PLD or PLD alone. PFS and OS were the co-primary endpoints. The combination group achieved a non significant longer OS (15.7 mos; HR vs. PLD 0.89, 95% CI, 0.74–1.24, *p* = 0.2) and PFS (3.7 mos, HR 0.78, 95% CI, 0.59–1.24; *p* = 0.03) with respect to PLD single agent. Avelumab alone compared to PLD did not improve either OS (HR = 1.14, 95% CI, 0.95–1.58; *p* = 0.82) or PFS (HR = 1.68, 95% CI, 1.32–2.60; *p* > 0.99). A longer PFS and OS trend was observed in the avelumab + PLD arm vs. PLD arm among the PD-L1+ patients (58%). In the avelumab, avelumab plus PLD, and PLD arms, ≥G3 TRAEs occurred in 49.7%, 68.7%, and 59.3% of patients, respectively [25].

In the phase II ENCORE-603 study, 126 patients with PR-ROC were randomized 2:1 to avelumab plus the class I selective histone deacetylase (HDAC) inhibitor entinostat or avelumab plus placebo (PBO), with PFS as the primary endpoint. No significant differences were detected between the two groups for mPFS (1.64 vs. 1.51 mos; HR = 0.9, 95% CI, 0.58–1.39; *p* = 0.031), ORR, and OS. AEs were more frequent with the avelumab–Entinostat combination than with avelumab–PBO (93% vs. 78%, ≥G3 AEs 41% vs. 10%), with fatigue, nausea, diarrhea, anemia, and chills being the most frequent [26].

#### 3.2.2. Atezolizumab

In the randomized phase III IMagyn050/GOG 3015/ENGOT-OV39 trial, 1,301 stage III/IV OC patients were treated with the combination of carboplatin, paclitaxel, and bevacizumab plus or minus atezolizumab (1200 mg q3w). The PFS improvement was not significant both in the overall population (19.5 vs. 18.4 mos HR = 0.92; 95% CI, 0.79–1.07; *p* = 0.28) and among the PD-L1 positive patients (20.8 vs. 18.5 mos HR = 0.8; 95% CI, 0.65–0.99; *p* = 0.38). Similarly, no benefit emerged at the interim analysis for OS. Anemia, neutropenia, and hypertension were the most common ≥G3 AEs [27].

#### 3.2.3. Durvalumab

Durvalumab was tested in combination with chemotherapy, PARPis, or anti-VEGF agents.

In the NCT02431559 phase I/II trial, 40 PR-ROC women received durvalumab plus PLD, reaching an ORR of 15% and a 6-mos PFS rate of 47.7%. The most frequent TRAEs were palmar–plantar erythrodysesthesia syndrome (PPES), stomatitis, fatigue, abdominal pain, nausea, fever. G3 TRAEs occurred in at least two patients and included lymphopenia, anemia, increased lipase, rash, and stomatitis [28].

The combination of durvalumab and olaparib was tested in three phase II studies. The MEDIOLA study aimed to evaluate 12 w DCR and safety as primary endpoints, plus 28 w DCR, ORR, DOR, PFS, and OS as secondary endpoints. Initially, 32 women with BRCA-mutant PS-ROC were included. The 12 w DCR was 81% [29]. After a median follow-up of 20.4 mos, 28 w DCR was 65.6% with mPFS 11.1 mos, ORR 71.9%, and mOS was not reached [30]. Subsequently, the study included 63 BRCA-wild type patients. Thirty-two patients received durvalumab plus olaparib, 31 patients were treated with olaparib plus durvalumab plus bevacizumab. The doublet cohort reached an ORR of 31.3% (95% CI 16.1–50.0%) vs. 77.4% in the triplet cohort (95% CI 58.9–90.4%). The mPFS was 5.5 mos for the doublet cohort and 14.7 mos for the triplet cohort, respectively. The 24 w DCR was 28.1% in the doublet cohort and 77.4% in the triplet cohort. The most common ≥G3 AEs were anemia, neutropenia, and lipase increased in both cohorts, while in the triplet cohort, hypertension and fatigue were also registered. Six percent and sixteen percent of patients discontinued the treatment in the double and triplet cohorts, respectively [31]. In the dose-escalation phase I/II NCT02484404 trial, among 35 patients with ROC, a DCR of 53% was observed with durvalumab plus olaparib or cediranib (5 PR, 13 stable disease [SD]). ≥G3 AEs included anemia (26%) and lymphopenia (14%) [32]. In a third single-center study (NCT02484404), 35 patients with PR-ROC were included. The (primary endpoint) ORR was 14%. Exploratory analyses showed that an increased gamma-interferon (γ-IFN) production was associated with longer PFS (*p* = 0.023), whereas increased vascular endothelial growth factor receptor (VEGFR)-3 levels determined shorter PFS (*p* = 0.017). Haematologic toxicity caused the highest ≥G3 AEs (most frequently anemia, 31%) [33].

### 3.3. Anti CTLA-4

Few trials have explored the activity of single agents anti-CTLA-4 Ipilimumab or Tremelimumab in the advanced/recurrent OC with unsatisfactory results. In the NCT01611558 phase II trial with ipilimumab at the dose of 10 mg/kg, 38 out of 40 PR-ROC patients did not complete treatment due to PD, severe toxicity, or death [34]. The combination with PARPis is still at an early stage but seems to be tolerated and induces anti-tumor responses. More specifically, 24 PR-ROC patients received tremelimumab alone or combined with olaparib in the NCT02485990 phase II trial, with 1 PR and 9 SD. No G4 AEs were reported, while the most common G3 toxicities were rash (13%), hepatitis, and colitis (both 8%) [35]. The same combination was administered to three BRCA-mutant OC patients in the NCT02571725 phase I trial, with a good safety profile (only G1/2 AEs were reported) and decreased tumor size after three cycles [36].

## 4. Discussion

Given the impact on morbidity and mortality among the female population, the search for new therapeutic options represents an unmet need for OC. Immunotherapy has revolutionized the treatment landscape of many solid tumors in the last ten years, and it now represents the first therapeutic approach with impressive survival benefits in diseases such as lung cancer, melanoma, renal cell carcinoma [10]. However, limited benefits have emerged in OC, even leading to premature termination due to the futility of some studies. Different components of the OC tumor microenvironment (TME) contribute to this failure, such as myeloid-derived suppressor cells (MDSCs), tumor-associated macrophages (TAMs), T-cells, cytokines, and soluble factors [37,38,39,40]. MDSCs exert immunosuppressive functions, such as the inhibition of T-effector and natural killer (NK)-cells, and are induced under pro-inflammatory cytokines, IFNγ, tumor necrosis factor-alpha (TNFα), interleukin (IL)-6 [41]. In OC, IL-6 plays a negative prognostic role and is associated with high MDSCs, and tumor progression [42,43]. The inflammatory cytokines cooperate to induce cyclooxygenase-2 (COX-2) and lead to prostaglandin E2 (PGE2) synthesis, which limits T-cell recruiting at tumor sites, together with VEGF [44,45]. TAMs are recruited at ovarian tumor sites, and IL-6, IL-10, transforming growth factor (TGF)-β promote their differentiation in M2 macrophages, associated with tumor invasiveness, spread, and angiogenesis [46,47,48]. M2 macrophages increase with the OC stage when contemporary M1 macrophages decrease, playing a negative prognostic role [49,50,51]. Moreover, they promote immunosuppression by producing cytokines (IL-1R, IL-10, C-C Motif Chemokine Ligand [CCL]17, CCL20, CCL22) that inhibit T-effectors proliferation and enhance Tregs function [52,53,54]. Treg cells are associated with advanced stages of OC and have a negative prognostic and immunosuppressive role [54]. They produce IL-10 and TGFβ, contributing to the inhibition of effector T-cells [55]. High levels of immunosuppressive elements within OC TME can also weaken dendritic cells and antigen-presenting cells (APCs) activity [56]. More accurate knowledge of the TME of the primary tumors and the metastatic sites will facilitate the design of more effective treatment combinations (Figure 1).

OC encompasses a heterogeneous group of malignancies that in over 95% of cases have an epithelial origin and are more frequently represented by high grade serous ovarian carcinoma (HGSOC) (70% of cases), followed by endometrioid ovarian cancer (EOC) (10%), clear cell OC (ccOC) (10%), low-grade serous OC (LGSOC, less than 5%), and mucinous OC (MOC, around 3%) [3]. Among them, the ccOC seems to be the most immunogenic: it more frequently carries the DNA microsatellite instability (MSI), has higher CD8+ tumor-infiltrating lymphocites (TILs), CD8+/CD4+ ratio, and higher PD-L1 levels [57,58]. Effectively, it is five times more responsive to ICIs than other OC subtypes [19]. Even among HGSOC, at least four different genomic classes were identified in The Cancer Genomic Atlas registry, differing for immunoreactivity. A unique subtype expresses genes related to immune sensitivity such as Toll-like receptor (TLR), TNF and is characterized by higher TILs infiltration [59,60]. Moreover, proteomics studies showed that the four subclasses of HSGOC are characterized by different expressions of proteins involved in DNA replication, ECM and cellular interaction, and cytokine signaling that contributes to immune responsiveness [61]. In our opinion, the different ICIs response observed among OC patients is rooted in the inter-tumor heterogeneity. Therefore, a deeper insight into the genomics characteristics of OC and their relationship with the immunological profile could allow us to better clarify the predictive factors for ICIs response. Ideally, specific immunogenomic scores could be developed for more accurate patients selection.

OC has been indicated as potentially more immune responsive when carrying BRCA mutations or homologous recombination deficiency (HRD). In fact, the impaired DNA repair leads to neo-antigens production, resulting in a higher tumor mutational burden (TMB) (even if <10 mutations per megabase are usually detected) and recruiting TILs at tumor sites. However, HRD or BRCA mutations were not linked to a higher sensitivity to ICIs in the IMagyn050 nor in the Javelin Ovarian 100 trials [23,27]. BRCA-mutant/HRD OC is associated with higher CD3+ and CD8+ TILs, PD1/PD-L1 levels, and genes related to cytotoxicity, such as T-Cell Receptor (TCR), γ-IFN, and TNF-Receptor pathway [62,63,64,65]. As proof of this, in the NCT02484404 trial, durvalumab plus olaparib determined a longer PFS in case of increased γ-IFN production [33]. Another mechanism of immune responsiveness is represented by the mismatch repair (MMR) deficiency, harboring the DNA MSI. MSI tumors produce neo-antigens, with a 10–100-fold higher TMB than MS stable (MSS)-tumors, resulting in high immunogenicity. Some genes triggering MSI were also identified in a percentage ranging from 17% to 59% of OC (more commonly in non-serous subtypes): the oncosuppressor TP53; Dihydropyrimidinase-related protein (DPYSL)-2, involved in microtubules function; Alpha Kinase (ALPK)-2, with a role in apoptosis and DNA repair [66]. In Lynch syndrome, a germline mutation of the MMR genes MutL homolog (MLH)-1, MutS homolog (MSH)-2 and -6, PMS1 homolog (PMS)-2 leads to an increased risk to develop some cancer subtypes, including OC [67]. Therefore, these tumors may be good candidates for ICIs treatment. Other genes could be involved in ICI response, justifying the different results observed among OC patients. The SWItch/Sucrose Non-Fermentable (SWI/SNF) complex consists of around 15 subunits, acting as a chromatin remodeler. In other tumor subtypes, the loss of function of the SWI/SNF complex predicts ICI response, increasing MMR deficiency, TMB, and neo-antigens production [68]. SWI/SNF complex mutations were frequently detected in OC [69]. We can assume that genetic diversity contributes to different ICI responses among OC patients. A more extensive genetic characterization could allow more accurate identification of responders and non-responders.

The possible relationship between platinum- and immunotherapy-sensitivity/resistance is also a field that merits further investigation [70]. A series of genetic and epigenetic elements were identified to drive platinum response: alterations of p53, specific microRNAs, elements driving the epithelial-to-mesenchymal transition (EMT), HRD, and BRCA mutations [71]. Since BRCA mutation and HRD were proposed to correlate with platinum sensitivity in contemporary deficient nucleotide excision repair, the co-administration of PARPis and ICIs in PS-ROC could result in higher ORR and survival rates [29,30,31]. PARPis enhance ICIs activity because they induce the release of neoantigens, increasing the TMB, promote PD-L1 expression, and directly activate the IFN genes; however, this was determined in OC [17,29,30,31,72,73]. Many ongoing trials are addressing this combination strategy in the advanced setting (Table 2).

As ICIs monotherapies showed only minimal results in terms of response rate and survival in OC, the combination with agents with different mechanisms of action appears a promising strategy to increase efficacy. Although chemotherapy represents a cornerstone in the treatment of advanced OC, it was historically perceived to play an immunosuppressive role. On the contrary, more recently, it has emerged that platinum derivatives promote APCs and their function, activating the immune response [74,75,76,77]. Doxorubicin plays an immunomodulatory effect, reducing the immunosuppressive state and improving tumor sensitivity to NK and CD8+ T-cells [78]. Low-dose cyclophosphamide also holds immunomodulatory properties, such as Tregs reduction and CD8+ cells induction [79,80]. However, the studies conducted so far did not lead to survival improvements. Besides the immunological potential, timing and schedule should be more deeply investigated and optimized for improving efficacy. The combination of ICIs and anti-VEGF agents seems attractive because the anti-angiogenic drugs directly influence OC TME [20,31,32,81,82,83]. Other combinations with multikinase inhibitors targeting VEGF/VEGFRpathway, such as cabozantinib or lenvatinib, are now under evaluation. The association with other agents with immunotherapeutics role, such as the anti-Lymphocyte-activation gene 3 (LAG-3) Relatlimab, as well as monoclonal antibodies such as the anti-Cluster of differentiation (CD)27 Varlilumab, the anti-CD47 Magrolimab, is under investigation (Table 2). Actually, overcoming the immunosuppressive pathways in the TME could represent a complementary way to potentiate ICIs effect on the immune system. Therapeutic vaccines were administered in OC, inducing cellular and humoral responses but rarely survival improvement as monotherapies [84]. Hence, several tumor-associated antigens were found in OC, such as p53, folate receptor (FR), New York Esophageal Squamous Cell Carcinoma-1 (NY-ESO-1), and Ca125 [85,86,87,88]. Therefore, combinations of ICIs and vaccines need to be explored. New approaches such as autologous TILs, cancer cell therapy, and adoptive cell therapy (ACT) also represent future possibilities for improving ICIs efficacy (Table 2).

Currently, a uniformly accepted predictive role of PD-L1 for ICIs response was not yet identified in solid tumors, including OC. PD-L1 expression varies between primary tumors and metastases, implying heterogeneity [89]. However, even if PD-L1 positivity was retrieved in around 1/3 OCs, the clinical impact was not elucidated, with conflicting results regarding the association with higher tumor stage/grade or shorter survival [90,91,92,93,94]. Indeed, some of the published trials reported better results for PD-L1 positive than PD-L1 negative patients [12,13,27]. In other studies, PD-L1 positivity was not predictive of ICIs response [19,20]. Recent research has focused on the post-transcriptional modifications of PD1, and even more PD-L1, which N-glycosylation of specific sites functionally modulates. PD-L1 and PD1 N-glycosylation ensure stability, prevents clearance, and influences mutual interactions [95,96]. The N-glycosylation of the PD1/PD-L1 receptors and its aberrations should be better investigated as possible immune resistance mechanisms in OC since specific glycoproteomic signatures were found in HGSOC: the immunoreactive subtype was richer in mannose than the mesenchymal, which was mainly fucosylated [97]. Moreover, it was evidenced that the antibodies used in the immunohistochemical analysis for PD-L1 accessed the highly glycosylated PD-L1 with difficulty, resulting in a certain percentage of PD-L1 false-negative results partially explaining ICIs efficacy also in PD-L1 negative patients [98]. More profound knowledge of the post-transcriptional status of PD1/PD-L1 and the search for biomarkers with a predictive role for ICIs’ efficacy is warranted to ensure the best patient selection.

## 5. Conclusions

Thus far, OC remains one of the few malignancies in which ICIs have not changed the standard of care, and neither monotherapies nor combinations have been approved. Effectively, significant heterogeneity was identified across OC patients at the genomic, proteomic, glycoproteomic, and immunologic levels, that in our opinion, should be further investigated to improve ICIs efficacy. We also believe that the combinations of ICIs with agents with different mechanisms of action will strengthen ICIs efficacy in OC. The combinations of ICIs with anti-VEGF agents or PARP-inhibitors represent potentially very effective associations, and several studies examine this strategy. However, schedule and timing should be optimized in order to preserve tolerability. Combinations with other agents, such as multikinase inhibitors, immunotherapies targeting the immunosuppressive network in the TME, or vaccines, should be further explored to maximize the efficacy with minimal toxicity.

Besides PD-L1, biomarkers with a predictive role to ICIs should be investigated. Integrating such biomarkers with genomic and immunologic profiling will provide a comprehensive understanding of OC, guiding clinical trials towards rational therapy combinations and sequencing.

## Figures and Tables

**Figure 1 cancers-13-04438-f001:**
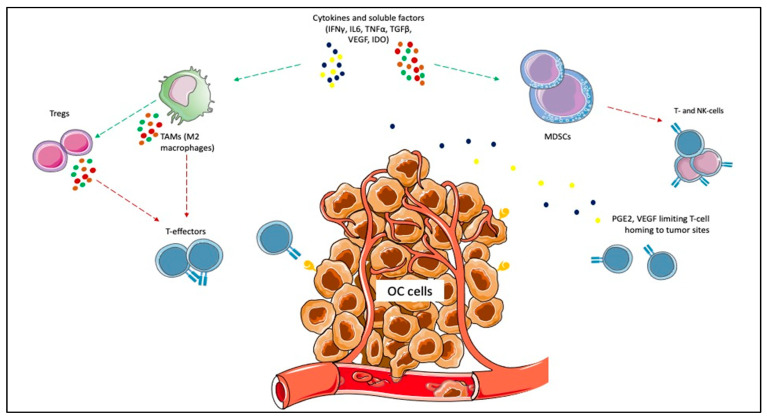
Immunosuppressive elements of ovarian cancer (OC) microenvironment. Cytokines and other soluble factors, such as interferon-gamma (IFNγ), tumor necrosis factor-alpha (TNFα), interleukin (IL)-6, IL-10, and transforming growth factor-beta (TGFβ) induce the proliferation of myeloid-derived suppressor cells (MDSCs) and the polarization of tumor-associated macrophages (TAMs) towards the M2 subtype. MDSCs exert immunosuppressive functions, such as the inhibition of T-effector and natural killer (NK)-cells. The inflammatory cytokines cooperate to induce cyclooxygenase-2 (COX-2) and lead to prostaglandin E2 (PGE2) synthesis, which limits T-cell recruiting at tumor sites. M2 macrophages promote immunosuppression by producing cytokines (e.g., IL-1R, IL-10, C-C Motif Chemokine Ligand [CCL]17, CCL20, CCL22) that inhibit T-effectors proliferation and enhance Tregs function.

**Table 1 cancers-13-04438-t001:** Trials of ICIs in OC.

Study Name	Phase	Target Population (Number of Pts)	Administered Drugs	Primary EP	Results
Keynote-100 (NCT02674061) [12,13]	II	Cohort A ROC after 1–3 therapies, TFI 3–12 mos (*n* = 285); Cohort B ROC after 6 lines, TFI >3 mos (*n* = 91)	Pembrolizumab	ORR (RECIST and by PD-L1)	ORR 8.0%, DC R 37% (overall), 1/3 responses > 6 mos; mDOR (A), NR (B); mPFS 2.1 mos; PD-L1+: ORR 17.1%, OS 21.9 (cohort A) and 24.0 mos (B); ≥G3 AEs: 19.7%; 2 treatment-related deaths.
Keynote-028 (NCT02054806) [14]	Ib	PD-L1+ ROC (*n* = 26)	Pembrolizumab	ORR	ORR 11.5%, mPFS 1.9 mos, mOS 13.8 mos AEs 73.1% (1 G3)
NCT02865811 [15]	II	PR-ROC, fallopian tube or peritoneal cancer (*n* = 26)	Pembrolizumab + PLD	Clinical Benefit Rate (CR, PR, SD)	ORR 19%, 3 PR, 1 SD > 24 w) ≥G3 AEs: rash (19%), ↑ALT (8%)
NCT02853318 [16]	II	PS- and PR-ROC (*n* = 40)	Pembrolizumab + bevacizumab + cyclophosphamide	ORR, mPFS	ORR 47.5%, mPFS 10 mos; 6 mos PFS: 100% (PS-ROC), 59% (PR-ROC) (*p* = 0.024) <G3 AEs: fatigue (45.0%), diarrhea (32.5%), hypertension (27.5%); ≥G3: hypertension (15.0%), lymphopenia (7.5%).
TOPACIO/Keynote-162 (NCT02657889) [17]	I–II	PR-ROC (*n* = 62)	Pembrolizumab + niraparib	ORR	ORR 25%, DCR 68%; BRCAm: ORR 45%, DCR 73% ≥G3 AEs: anemia (21%), thrombocytopenia (9%).
UMIN000005714 [18]	II	PR-ROC (*n* = 20)	Nivolumab (1 and 3 mg/kg)	BOR	2 CR; DCR 45%; mPFS 3.5 mos; mOS 20 mos ≥G3 AEs 40%, 2 SAEs, 11% discontinuation.
NRG-GY003 (NCT02498600) [19]	II	PS- and PR-ROC (*n* = 100)	Nivolumab vs. nivolumab + ipilimumab → nivolumab maintenance	ORR	ORR 31.4% (N + I) vs. 12.2% (N) (*p* = 0.034); PFS 3.9 (N + I) vs. 2 (N) mos (HR = 0.53); OS 28.1 (N + I) vs. 21.8 (N) mos (HR = 0.79); responses not associated with PD-L1 ≥G3 AEs: 33% (N), 49% (N + I).
NCT02873962 [20]	II	PS- and PR-EOC (*n* = 38)	Nivolumab + bevacizumab	ORR	ORR: 28.9% (40% PS-ROC, PR-ROC 16.7%); mPFS 9.4 mos (12.1 mos PS-ROC); PD-L1- better than PD-L1+ pts. AEs 89.5%: fatigue (47.4%), headache (28.9%), myalgia (28.9%), ↑amylase (28.9%), ↑AST (26.3%), hypertension (26.3%); pneumonitis (10.5%), colitis (5.3%). ≥G3 AEs 23.7%.
NINJA [21]	III	PR-ROC (Japanese population, *n* = 316)	Nivolumab vs. gemcitabine or PLD	OS	No OS differences (HR = 1.03); PFS 2 (N) vs. 3.8 mos (gem/PLD) (HR = 1.46; *p* = 0.002) ≥G3 AEs: 22.4% (N), 68.4% (gem/PLD)
JAVELIN (NCT01772004) [22]	Ib	PR-ROC (*n* = 125)	Avelumab	BOR	1/125 CR, 11/125 PR; 1 yr PFS: 10.2%; mOS: 11.2 mos; mPFS: 2.6 mos ≥G3 AEs 7.2% (↑lipase 2.4%)
JAVELIN 200 (NCT02580058) [25]	III	PR-ROC (*n* = 566)	Avelumab vs. avelumab + PLD vs. PLD	PFS, OS	Ave + PLD: PFS 3.7 mos (HR vs. PLD = 0.78, *p* = 0.03), OS 15.7 mos (HR vs. PLD = 0.89, *p* = 0.2); avelumab vs. PLD HR for OS = 1.14, HR for PFS = 1.68 PD-L1+: trend for longer PFS and OS Ave+PLD vs. PLD ≥G3 AEs: 49.7% (Ave), 68.7% (Ave + PLD), 59.3% (PLD)
ENCORE-603 (NCT02915523) [26]	II	PR-ROC (*n* = 126)	Avelumab + Entinostat vs. avelumab + PBO	PFS	mPFS = 1.64 (A + E) vs. 1.51 mos (A + P) (*p* = 0.031). No differences in ORR (6% vs. 5%), or OS (NE vs. 11.3 mos) AEs: 93% (A + E), 78% (A + P); ≥G3 AEs: 41% (A + E), 10% (A + P)
IMagyn050 (NCT03038100) [27]	III	First-line OC (*n* = 1301)	CHT (CBDCA + paclitaxel) + bevacizumab + atezolizumab vs. CHT + beva + PBO	PFS/OS in ITT and PD-L1+ population	PFS 19.5 vs. 18.4 mos (HR = 0.92; *p* = 0.28); PD-L1 + PFS: 20.8 vs. 18.5 mos (HR = 0.8; *p* = 0.38); no OS advantage. ≥G3: neutropenia, hypertension, anemia.
NCT02431559 [28]	I–II	PR-ROC (*n* = 40)	Durvalumab + PLD	PFS6	PFS6: 47.7%; ORR 15% G3 Aes in ≥2 pts: lymphopenia, anemia, lipase increased, rash, stomatitis
MEDIOLA (NCT02734004) [29,30,31]	II	gBRCAm (*n* = 32) and BRCAwt (*n* = 63) PS-ROC	gBRCAm group: olaparib (4 w) → durvalumab + olaparib BRCAwt group: durvalumab + olaparib (D + O; *n* = 32), durvalumab + olaparib + bevacizumab (D + O + B; *n* = 31)	12 w DCR, safety	gBRCAm group: 12 w DCR 81%, mPFS 11.1 mos, ORR 71.9% BRCAwt D + O group: ORR 31.3%, mPFS 5.5 mos, 24 w DCR 28.1%; 6% discontinuation BRCAwt D + O + B group: ORR 77.4%, mPFS 14.7 mos, 24 w DCR 77.4%; 16% discontinuation ≥G3 AEs: anemia, lipase increased, neutropenia; + hypertension, fatigue (O + D + B cohort)
NCT02484404 [32]	I–II	PS/PR-ROC (*n* = 35)	Durvalumab + olaparib or cediranib	RP2D	5 PR, 13 SD, DCR 53% ≥G3 AEs: anemia (26%), lymphopenia (14%)
NCT02484404 [33]	II	PR-ROC (*n* = 35)	Durvalumab + olaparib	ORR	ORR 14%; longer PFS with ↑IFNγ (*p* = 0.023), shorter PFS with ↑VEGFR3 (*p* = 0.017) ≥G3 AE: anemia (31%).
NCT02485990 [35]	I–II	PR-ROC (*n* = 24)	Tremelimumab vs. tremelimumab + olaparib	RP2D, ORR	1 PR, 9 SD ≥G3 AEs: rash (13%), hepatitis (8%), colitis (8%); no ≥ G4 AEs.
NCT02571725 [36]	I–II	BRCAm OC (*n* = 3)	Tremelimumab + olaparib	RP2D	G1/2 AEs, decreased tumor size after 3 cycles

AE: adverse event; ALT: alanine aminotransferase; AST: aspartate aminotransferase; BOR: best overall response; BRCAwt: BRCA-wild type; CBDCA: carboplatin; CHT: chemotherapy; CR: complete response; DCR: disease control rate; DOR: duration of response; EOC: epithelial ovarian cancer; EP: endpoint; gBRCAm: germline BRCA-mutated; ICI: immune-checkpoint inhibitor; ITT: intention-to-treat; mOS: median overall survival; mPFS: median progression-free survival; OC: ovarian cancer; ORR: overall response rate; PBO: placebo; PD-L1: programmed death ligand 1; PFS6: 6-months progression-free survival; PLD: pegylated liposomal doxorubicin; PR: partial response; PR-ROC: platinum-resistant recurrent ovarian cancer; PS-ROC: platinum-sensitive recurrent ovarian cancer; RP2D: recommended phase II dose; SD: stable disease; TFI: treatment-free interval; VEGFR: vascular endothelial growth factor receptor; →: followed by; ↑: increased.

**Table 2 cancers-13-04438-t002:** Ongoing trials of ICIs combinations in the metastatic OC.

Clinicaltrials.gov Registration Number (Name)	Phase	ICI Combinations (Drug Class)
NCT03508570	I	Nivolumab + Ipilimumab
NCT03355976	II	
NCT02834013	II	
NCT03959761	I–II	Nivolumab (IP) + Surgery plus HIPEC
NCT02737787	I	Nivolumab + WT1 or NY-ESO-1 (vaccine)
NCT03522246 (ATHENA)	III	Nivolumab + Rucaparib (PARPi)
NCT02873962	II	Nivolumab + Bevacizumab (anti-VEGF) ± Rucaparib
NCT04611126	I–II	Nivolumab + Relatimab (anti-LAG-3) + Ipilimumab + ACT
NCT03100006	IB-IIA	Nivolumab + Oregovomab (anti-Ca125)
NCT04620954 (ORION-02)	I–II	Nivolumab + Oregovomab + PLD + CBDCA
NCT03667716	I	Nivolumab + COM701 (PVRIG inhibitor)
NCT04570839	I–II	Nivolumab + COM701 + BMS-986207 (anti-TIGIT)
NCT04514484	I	Nivolumab + Cabozantinib (TKI)
NCT02335918	I–II	Nivolumab, Varlilumab (anti-CD27)
NCT02526017	I	Nivolumab + Cabiralizumab (anti-CSF1R)
NCT02440425	II	Pembrolizumab + Paclitaxel
NCT03029598	I–II	Pembrolizumab + CBDCA
NCT04387227	II	
NCT04575961 (PERCEPTION)	II	Pembrolizumab + Platinum-based CTx
NCT02766582	II	Pembrolizumab + CBDCA + Paclitaxel
NCT02834975	II	
NCT03410784 (MITO28MaNGOov4)	II	
NCT02520154	II	
NCT03126812	I–II	
NCT03539328	II	Pembrolizumab + Gemcitabine or Paclitaxel or PLD vs. CTx
NCT02900560	II	Pembrolizumab ± Azacitidine
NCT02901899	II	Pembrolizumab + Guadecitabine
NCT03596281 (PEMBOV)	I	Pembrolizumab + Bevacizumab + PLD
NCT04417192 (OLAPem)	II	Pembrolizumab + Olaparib
NCT03740165 (MK-7339-001/KEYLYNK-001/ENGOT-ov43/GOG-3036)	III	CBDCA + Paclitaxel → Pembrolizumab + Olaparib vs. Pembrolizumab + PBO vs. PBO + Olaparib
NCT04519151	II	Pembrolizumab + Lenvatinib
NCT03797326 (MK-7902-005/E7080-G000-224/LEAP-005)	II	
NCT04781088	II	Pembrolizumab + Lenvatinib + Paclitaxel
NCT02606305	I–II	Pembrolizumab + Mirvetuximab soravtansine (anti-FRα ADC)
NCT03734692	I–II	Pembrolizumab + IP Rintatolimod (anti-TLR3) + Cisplatin
NCT03158935 (ACTIVATE)		Pembrolizumab + Cyclophosphamide + autologous TILs + IL-2
NCT03029403	II	Pembrolizumab + DPX-Survivac (vaccine) + Cyclophosphamide
NCT03113487	II	Pembrolizumab + p53 MVA (vaccine)
NCT04713514 (TEDOVA)	II	Pembrolizumab + OSE2101 vs. OSE2101 (multi-epitope vaccine)
NCT03558139	I	Avelumab + Magrolimab (anti-CD47)
NCT04510584	II	Atezolizumab + Bevacizumab
NCT02891824 (ATALANTE)	III	Atezolizumab + Bevacizumab + platinum-based Ctx vs. PBO + Bevacizumab + platinum-based Ctx
NCT03353831	III	Atezolizumab + Bevacizumab + Ctx vs. Bevacizumab + Ctx
NCT02839707	II–III	Atezolizumab + Bevacizumab + PLD
NCT02659384	II	Atezolizumab + Bevacizumab ± acetylsalicylic acid
NCT03363867 (BEACON)	II	Atezolizumab + Bevacizumab + Cobimetinib
NCT03695380	I	Atezolizumab + Cobimetinib (anti-MEK) + Niraparib
NCT03598270 (ANITA)	III	Atezolizumab + Platinum-based Ctx vs. platinum-based Ctx → Niraparib ± Atezolizumab maintenance
NCT02914470 (PROLOG)	I	Atezolizumab + CBDCA + Cyclophosphamide
NCT03206047	I–II	Atezolizumab + Guadecitabine + CDX-1401 (vaccine)
NCT03073525	II	Atezolizumab + Vigil (cancer cell therapy)
NCT01975831	I	Durvalumab + Tremelimumab
NCT02953457	II	
NCT03026062	II	
NCT04644289 (WoO)	II	Durvalumab + Olaparib
NCT04742075 (DOVACC)	II	Durvalumab + Olaparib + UV-1
NCT04015739 (BOLD)	II	Durvalumab + Olaparib + Bevacizumab
NCT03737643 (DUO-O)	III	Durvalumab + platinum-based Ctx + Bevazicumab vs. PBO + platinum-based Ctx + Bevacizumab → Durvalumab + Bevacizumab + Olaparib maintenance
NCT03699449 (AMBITION)	II	Durvalumab + Tremelimumab or Olaparib or Cediranib or Ctx
NCT02726997 (N-Dur)	I–II	Durvalumab + CBDCA + Paclitaxel
NCT03430518	I	Durvalumab + Eribuline
NCT03085225 (TRAMUNE)	I	Durvalumab + Trabectedin
NCT02811497 (METADUR)	II	Durvalumab + Azacitidine
NCT02764333	II	Durvalumab + TPIV200 (anti-FR vaccine)
NCT02725489	II	Durvalumab + Vigil
NCT03267589	II	Durvalumab + Tremelimumab + MEDI 9447 (anti-CD73 Ab) + MEDI 0562 (anti-OX40)
NCT04019288	I–II	Durvalumab + AVB-S6-500 (Anti-AXL Fusion Protein)
NCT03277482	I	Durvalumab + Tremelimumab + RT
NCT02571725	II	Tremelimumab + Olaparib
NCT03602859	III	Dostarlimab (anti-PD1) + Ctx vs. Ctx + Niraparib vs. Ctx + PBO

ACT: adoptive cell therapy; ADC: antibody-drug conjugate; CBDCA: carboplatin; CD: cluster of differentiation; CSF1R: Colony-stimulating factor 1 receptor; FRα: folate receptor alpha; HIPEC: Hyperthermic intraperitoneal chemotherapy; IL-2: Interleukin-2; IP: intra-peritoneal; LAG-3: Lymphocyte-activation gene 3; MVA: Modified Vaccinia Virus Ankara; NY-ESO-1: New York Esophageal Squamous Cell Carcinoma-1; PBO: placebo; PLD: pegylated liposomal doxorubicin; PVRIG: poliovirus receptor-related immunoglobulin domain containing; RT: Radiation Therapy; TIGIT: T-cell immunoreceptor with immunoglobulin and immunoreceptor tyrosine-based inhibitory motif; TIL: tumor-infiltrating lymphocytes; TKI: tyrosine-kinase inhibitor; TLR3: Toll-Like Receptor 3; VEGF: vascular-endothelial growth factor; WT1: Wilms tumor 1; →: followed by.

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
