# Peer review of "Ovarian Cancer in the Era of Immune Checkpoint Inhibitors: State of the Art and Future Perspectives"

_cancers, 2021, doi:10.3390/cancers13174438_

Round 1

Reviewer 1 Report

This review is a nice summary on the clinical trials conducted with immune checkpoint inhibitors in advanced ovarian cancer either as a monotherapy or in combination with other therapies. The summary of the clinical trials is detailed and well described. 

Major Issue:

There is an inconsistence in the abstract and the review 

Abstract: "20 studies were selected, of which 16 were phase I or II and 4 phase III trials."

Results: "17 phase I or II and 3 phase III trials have been published (Table 1)."

Please correct as appropriate

Minor issues:

  1. Ovarian cancer should not be preceded by "the"
  2. Other grammatical changes are marked in the manuscript.

Author Response

We are extremely thankful to the reviewer for the precious comments and suggestions. We corrected the numeric inconsistency between the abstract and the review and modified the grammatical changes as suggested.

Reviewer 2 Report

This is a well organized (with use of tables and figure) and comprehensive review on such an important topic. Why are they not working? I feel the manuscript is ready for publication but would like to see the authors comment on uniqueness of ovarian cancer patients not responding? and why? in their thoughts etc. in particular there is alot of talk about glycosylation of the PD proteins and ovarian cancer associated with lots of glycosylation?? thoughts? The points in the discussion are all important points and appropriately based on current literature. 

Author Response

We are very thankful to the reviewer for the kind and useful comments. As suggested, we added some findings of genetics, proteomics, relationship with immunogenomics, and also regarding the glycosylation of PD1 proteins, aiming to address the inter-tumor heterogeneity, trying to find out possible explanations for the different ICI responses reported in OC patients.